# Physical Activity Determinants in Older German Adults at Increased Dementia Risk with Multimorbidity: Baseline Results of the AgeWell.de Study

**DOI:** 10.3390/ijerph19063164

**Published:** 2022-03-08

**Authors:** Maria Isabel Cardona, Marina Weißenborn, Isabel Zöllinger, Eric Sven Kroeber, Alexander Bauer, Melanie Luppa, Alexander Pabst, David Czock, Hans-Helmut König, Birgitt Wiese, Jochen Gensichen, Thomas Frese, Hanna Kaduszkiewicz, Wolfgang Hoffmann, Steffi G. Riedel-Heller, Jochen René Thyrian

**Affiliations:** 1German Center for Neurodegenerative Diseases (DZNE), Site Rostock/Greifswald, 17489 Greifswald, Germany; wolfgang.hoffmann@uni-greifswald.de (W.H.); rene.thyrian@dzne.de (J.R.T.); 2Department of Clinical Pharmacology and Pharmacoepidemiology, University Hospital Heidelberg, 69120 Heidelberg, Germany; marina.weissenborn@med.uni-heidelberg.de (M.W.); david.czock@med.uni-heidelberg.de (D.C.); 3Institute of General Practice and Family Medicine, University Hospital of LMU Munich, 80336 Munich, Germany; isabel.zoellinger@med.uni-muenchen.de (I.Z.); jochen.gensichen@med.uni-muenchen.de (J.G.); 4Institute of General Practice and Family Medicine, Martin-Luther-University Halle-Wittenberg, 06112 Halle (Saale), Germany; eric.kroeber@posteo.de (E.S.K.); alexander.bauer@medizin.uni-halle.de (A.B.); thomas.frese@uk-halle.de (T.F.); 5Institute of Social Medicine, Occupational Health and Public Health (ISAP), Medical Faculty, University of Leipzig, 04103 Leipzig, Germany; melanie.luppa@medizin.uni-leipzig.de (M.L.); alexander.pabst@medizin.uni-leipzig.de (A.P.); steffi.riedel-heller@medizin.uni-leipzig.de (S.G.R.-H.); 6Department of Health Economics and Health Services Research, University Medical Center Hamburg-Eppendorf, 20246 Hamburg, Germany; h.koenig@uke.de; 7Work Group Medical Statistics and IT-Infrastructure, Institute for General Practice, Hannover Medical School, 30625 Hannover, Germany; wiese.birgitt@mh-hannover.de; 8Institute of General Practice, University of Kiel, 24105 Kiel, Germany; kaduszkiewicz@allgemeinmedizin.uni-kiel.de; 9Institute for Community Medicine, University Medicine Greifswald, 17489 Greifswald, Germany

**Keywords:** multimorbidity, dementia, physical activity, determinants

## Abstract

Background: Multimorbidity is a common issue in aging societies and is usually associated with dementia in older people. Physical activity (PA) may be a beneficial nonpharmacological strategy for patients with complex health needs. However, insufficient PA is predominant in this population. Thus, there is an evident need to expand the knowledge on potential determinants influencing PA engagement among elderly persons at risk of dementia and multimorbidity. Methods: We used baseline data from the multicenter, cluster-randomized controlled AgeWell.de study. The main aim was to describe PA engagement and identify potential PA determinants in a sample of community-dwelling Germans aged 60–77 years old with an increased risk of dementia and multimorbidity. Results: Of the 1030 included participants, approximately half (51.8%) engaged in PA ≥2 times/week for at least 30 min at baseline. We identified self-efficacy (beta = 0.202, (*p* < 0.001) and BMI (beta = −0.055, (*p* < 0.001) as potential PA determinants. Conclusions: The identified determinants, self-efficacy, and BMI are consistent with those reported in the literature. Specific knowledge on PA determinants and stages of change in persons with risk of dementia and multimorbidity might guide the development of effective future prevention measures and health services tailored to this population. Trial registration: German Clinical Trials Register (reference number: DRKS00013555).

## 1. Introduction

In developed countries such as Germany, demographic aging has changed disease trends for delayed degenerative diseases and the presence of multiple health problems, also known as multimorbidity [1,2]. Multimorbidity is commonly defined as the presence of two or more chronic diseases or conditions. A recent cross-sectional study in Germany showed that of 123,224 participants, 62.1% of those aged 65 years and older were multimorbid, as measured by the presence of three chronic diseases from the 46-item checklist [3]. Moreover, multimorbidity is often associated with dementia in the elderly population [4]. On average, patients with dementia aged 65 years or older suffer from 4.6 accompanying chronic diseases [5]. The coexistence of different long-term chronic conditions has many implications for affected individuals since multimorbidity is strongly associated with poor clinical outcomes, such as reduced quality of life, disability, and high mortality risk, as well as with economic repercussions for health care systems due to increased hospital admissions, higher health care utilization, and reorganization of health services [1,4].

Evidence suggests that regular physical activity (PA) is a favorable nonpharmacological intervention in patients with complex health needs, including multimorbidity and dementia [6,7]. PA is understood as “any bodily movement produced by skeletal muscles that require energy expenditure above and beyond resting energy expenditure. It can be undertaken in many different ways: walking, cycling, sports and active forms of recreation (such as dance, yoga, and tai chi). PA can also be undertaken as part of work and as part of paid or unpaid domestic tasks around the home” [8]. In patients with complex health needs, PA is associated with improved health outcomes such as enhanced physical and cognitive function, activities of daily living (ADLs), and behavioral symptoms [6,9,10]. According to the World Health Organization (WHO), to achieve these positive health outcomes, older adults 65 years and above have to engage regularly in at least moderate-intensity aerobic PA (e.g., cardiorespiratory exercises that increase breathing and heart rate, such as brisk walking, cycling, or swimming) for 150 min/week [11]. In cases where older adults’ health conditions interfere with the previous recommendation, they should be as active as their current condition allows [12,13].

While regular PA is beneficial in many ways, PA engagement tends to decline with age. It is associated with increased physical exertion, resulting in avoidance of PA [12]. For instance, older adults engage in sedentary behavior (SB) for 65–80% (mean: 9.4 h/day) of their waking day [14,15]. SB is defined as any waking activity resulting in an energy expenditure of ≤1.5 metabolic equivalents (METs) while sitting, lying down, or resting [16]. In particular, individuals with ≥4 chronic physical conditions were found to have 1.45 times greater odds of high SB and higher mean min/day of SB [17]. Similarly, patients with multimorbidity and dementia tend to present more SB and lower PA levels than older adults who are healthy [18,19]. There is a growing body of evidence showing that SB is linked to a broad spectrum of negative health outcomes, such as physical functioning, diabetes, coronary heart disease, and premature death [20]. These patterns of SB and their negative impact on health, including increased risk of all-cause mortality, metabolic syndrome, large waist circumference, and overweightness/obesity [21], highlight the need to understand the determinants responsible for inactivity [12,22].

Over the last few decades, research on PA determinants has proliferated, with a focus on healthy elderly adults [12,23]. The evidence suggests that age, sex, health status, self-efficacy, and motivation are associated with PA [22]. Other studies have reported that obesity (BMI > 30 kg/m^2^), lower education, widowed or divorced marital status, and current smoking have a negative influence on PA engagement [24,25,26,27,28]. In particular, multimorbidly inactive participants were found to have significantly higher odds of being overweight/obese and lower levels of health-related quality of life [29]. However, evidence on PA determinants among elderly adults with multimorbidity and risk of dementia is still missing.

Several theories have emerged that focus on the psychological and psychosocial aspects of continuous and unidirectional models to explain the relationships between different predictors of PA [30]. Stage-based models such as the transtheoretical model (TTM) suggest that successful behavioral change requires movement between different stages of change and corresponding moderators such as decisional balance and self-efficacy [30]. According to the TTM, the PA stages of change are as follows: (a) precontemplation, which is characterized by low or no PA; (b) contemplation, in which there is low or no PA but performance of PA is considered in the future; (c) preparation, which is characterized by small changes in PA; (d) action, in which the person is physically active but still experiences difficulty; and e) maintenance, in which the person is physically active and finds it easy to maintain adherence [16]. A major moderator is self-efficacy, which is conceptualized as a person’s belief in his or her own abilities to successfully perform an action he or she proposes to perform [31]. Before people start to navigate between the stages of change, they begin to believe in their ability [32,33]. One study demonstrated that self-efficacy increased PA engagement in patients who had osteoarthritis in the knee [34]. Another study showed a positive correlation between self-efficacy and physical function in the performance of tai chi [35]. Thus, self-efficacy appears to be one of the major factors for initiating new behaviors, such as engagement in PA and long-term maintenance [32,33].

Few studies have investigated the existing theories on PA determinants in older adults with multimorbidity; therefore, potential determinants in this population are still unknown [36]. This study aimed to describe PA engagement based on the TTM stages of change and self-efficacy [37], determine its relationship to other patient characteristics, and identify potential PA determinants in a sample of community-dwelling Germans aged between 60 and 77 years at increased dementia risk with multimorbidity. Increased knowledge of PA determinants and stages of change in persons with dementia and multimorbidity might support the development of effective future prevention measures and health services adjusted to particular needs of this population [36].

## 2. Materials and Methods

### 2.1. Study Design

Using baseline data from the AgeWell.de study, the present research aimed to provide a descriptive analysis of sociodemographic and health-related lifestyle factors and PA engagement in a sample of community-dwelling Germans aged between 60 and 77 years old at increased dementia risk with multimorbidity. Moreover, we identified potential PA-related determinants in this population. AgeWell.de is a two-armed multicentric, cluster-randomized controlled trial (RCT, conducted in rural and urban sites (Leipzig, Kiel, Greifswald, Munich, and Halle)) in Germany. The aim is to deliver a multicomponent lifestyle intervention to primary care patients at increased dementia risk, targeting modifiable health and lifestyle factors of dementia over a period of 2 years (intervention group). The control group received general health advice/treatment as usual. The primary outcome is cognitive function. Secondary outcomes are measures of mental and physical health. The AgeWell.de trial is registered in the German Clinical Trials Register (DRKS; trial identifier: DRKS00013555). A detailed study protocol was published [38]; the study is currently ongoing.

### 2.2. Recruitment

The recruitment process and sample selection are displayed in Figure 1. Participants were recruited from among patients of community-dwelling general practitioners (GPs) between June 2018 and October 2019. The inclusion criteria were age between 60 and 77 years and an increased risk of dementia, which was quantified by the Cardiovascular Risk Factors, Aging, and Incidence of Dementia (CAIDE) score [39] with a cutoff of 9 points, as this score predicted dementia risk with a sensitivity of 0.77 and a specificity of 0.63 in previous studies [39]. The CAIDE score is based on information that is easy to assess (age, education, gender, blood pressure, body mass index, total cholesterol, and PA) and thus facilitates feasible case findings for eligible participants for GPs. The exclusion criteria were dementia diagnosed or suspected by the GPs; medical conditions potentially affecting safe engagement in the intervention (malignant disease/fatal illness, severe clinical depression, symptomatic cardiovascular disease, revascularization within the previous year) as judged by the GPs; severe loss of vision, hearing, or communicative ability/insufficient ability to speak and read German; severe mobility impairment; and coincident participation in another intervention trial.

### 2.3. Baseline Assessment

#### 2.3.1. Sociodemographic and Health-Related Factors

Initially, structured face-to-face interviews were conducted by GPs with all participants at baseline, assessing potential PA determinants, including sociodemographic information, lifestyle, and health-related factors. GPs provided further information on participants’ sociodemographic information, medical diagnoses, health-related lifestyle factors, laboratory data, and medications using standardized questionnaires and measures:the Montreal Cognitive Assessment (MoCA) [40] to assess cognitive performance;the Barthel Index [28] and the Amsterdam IADL scale [41] to identify ADLs and instrumental ADLs (IADLs), respectively;the Geriatric Depression Scale [42] to assess depressive symptoms;the Lubben Social Network Scale/LSNS-6 [43] to assess social isolation;anthropometric measures (height, weight, body mass index);a validated food frequency questionnaire [44] to collect information on nutrition, assessing volume and frequency of consumption of specific foods and beverages, including information on consumption of fruit, vegetables, and alcohol.

#### 2.3.2. PA Engagement

To assess PA engagement, we developed a questionnaire adapted from Lippke et al. [45]. We added a dichotomous variable asking individuals if they performed PA at least twice per week for at least 30 min. Furthermore, we included an item from the “Fit im Nordwesten” study led by Voelcker-Rehage addressing the TTM stages of change [46]. The five possible answers were as follows: (1) precontemplation stage: “I do not engage in PA once a week or more often at least 30 min, and I have no intention to do so.”; (2) contemplation stage: “I do not engage in PA once a week or more often at least 30 min, but I think about it.”; (3) preparation stage: “I do not engage in PA once a week or more often at least 30 min, but I have every intention to do so.”; (4) maintenance stage: “I engage in PA once a week or more often for at least 30 min, but I find it very hard.”; and (5) action stage: “I engage in PA once a week or more for at least 30 min, and it is easy.” Self-efficacy for PA was measured using health-specific self-efficacy scales proposed by Schwarzer and Renner [47]. Possible responses were (1) very uncertain, (2) somewhat uncertain, (3) somewhat certain, and (4) very certain. Moreover, we asked what types of PA (*n* = 7 types) participants engaged in, with respective frequency and duration of activity. The frequency was recorded using a five-level response format with levels ranging from “(almost) every day” to “rarely or never” first designed by Fuchs [48,49]. All necessary permissions for use in the trial were obtained from the respective copyright owners.

### 2.4. Statistical Analyses

Descriptive statistics were used to describe sociodemographic characteristics, cognitive status, GP diagnoses, health and lifestyle factors, PA engagement, stages of change for PA, self-efficacy, prevalence of PA types, and characteristics of patients according to participants’ stages of PA involvement in the sample at baseline. To test for differences between subgroups, we used Pearson’s chi-squared test and Welch’s t test. A *p* value of 0.05 was determined to indicate significant differences.

To identify potential PA determinants, logistic regression models were calculated in two steps. First, bivariate correlations between the dependent and independent variables were calculated with Pearson’s chi-squared test. In addition, the variables that showed a significant correlation with the outcome in the bivariate analysis (*p* < 0.05) were analyzed in a logistic regression procedure. Second, logistic regression was performed, and 95% confidence intervals (CIs) were estimated. Three models tested (1) psychological factors, (2) sociodemographic characteristics, and (3) health and lifestyle factors. The dependent variable for the regression models was regular PA engagement (i.e., PA at least two times/week for 30 min: yes/no), which is linked with positive health outcomes [11]. The independent variables tested in the regression models were perceived PA benefits, self-efficacy, cognitive status, sex, age, educational level, income in euros, marital status, BMI, depression, social network, ADLs, current smoking, alcohol consumption, and fruit and vegetable consumption. To identify which variables were affected with multicollinearity, we tested the variance inflation factors (VIFs). VIFs were less than two, suggesting a moderate correlation, although not sufficiently strong to justify any remedial action [50]. Data processing and statistical calculations were conducted using IBM SPSS version 25 (1989–2018 by IBM Corp.©, Armonk, New York, NY, USA).

### 2.5. Ethics

The study was approved by the responsible ethics committees of the coordinating center (Ethics Committee of the Medical Faculty of the University of Leipzig; ethical vote number: 369/17-ek) and of all participating study sites and expertise centers. Participants provided written informed consent to participate.

## 3. Results

### 3.1. Participants

We included 1.030 participants. A slight majority (*n* = 537) of participants were women. The mean age of the participants was 69.0 (SD = 4.9), and approximately two-thirds (64.6%) of the study sample was married/cohabiting. Approximately half had a self-reported intermediate education (55.2%). Regarding cognitive status, 57.7% of participants showed mild impairment. Many participants had been diagnosed with hypercholesterolemia/hyperlipidemia (71.4%), hypertension (87.0%), and obesity (BMI > 30 kg/m^2^) (54.4%). The mean scores were 1.6 (SD = 2.0) on the Geriatric Depression Scale, 20.76 (SD = 6.5) on the Lubben Social Network Scale, and 99.5 (SD = 3.1) for ADLs.

Concerning the lifestyles of the participants, 12.3% were current smokers, 60.0% drank alcohol once a week or more, and 48.6% consumed five portions of fruits and vegetables daily. Full sociodemographic characteristics, health and lifestyle factors, GP diagnoses, and motivation for PA of the participants are presented in Table 1.

### 3.2. PA Engagement

Half of the participants (51.8%) reported that they engaged in PA at least two times/week for at least 30 min. Reporting on the TTM stages of change for PA engagement, 20.0% of participants stated that they were in the precontemplation stage for PA, 19.0% were in the contemplation stage, 9.0% were in the preparation stage, 5.1% were in the maintenance stage, and the largest group (46.0%) was in the action stage. Regarding self-efficacy [47], 77.1% of participants were confident about exercising when they felt worried or had problems, as well as when they felt low (67.8%), tense (72.8%), tired (49.2%), or busy (45.2%) (Table 2).

### 3.3. Prevalence of PA Types

The most frequent types of PA performed by the participants were gardening and house chores (CI 95% [3.03, 3.19]); walking > 10 min (CI 95% [2.35, 2.52]); cycling (CI 95% [1.13; 1.31]); and gymnastics (CI 95% [1.00; 1.17]). Women more often practiced gardening and house chores (*p* = 0.006) and walking (*p* = 0.002), while men more often engaged in cycling (*p* = 0.013) (Table 3).

### 3.4. Characteristics of Patients According to the TTM Stages of PA Engagement

Participants in the precontemplation, contemplation, or preparation stages of PA engagement had a mean age of 68.8 years (SD = 4.9) and more often reported an intermediate level of education (59.0%), BMI ≥ 30 kg/m^2^ (60.4%), a diagnosis of hypercholesterolemia/hyperlipidemia (75.2%), and hypertension (87.9%). Those who reported being in the stage of maintenance or action had a mean age of 69.0 years (SD = 4.9); approximately two-thirds lived with a partner (67.8%), and 50.1% reported daily consumption of five portions of fruits and vegetables. The only significant group differences were found in BMI ≥ 30 kg/m^2^ (*p* = 0.001) (Table 4).

### 3.5. Potential PA Determinants

Table 5 presents the results of regression analysis for PA engagement of at least two times/week for 30 min as the dependent variable. In the first regression model, self-efficacy and cognitive status were the independent variables. This model described 18.3% of the sample and showed a significant positive significant correlation between PA engagement and self-efficacy (*p* < 0.001). The second model described 19.6% of the sample. Sex, age, educational level, and income were positively associated with PA engagement. This suggests that women, older participants, and those with higher educational levels and incomes more often engaged in PA. However, the relations between these variables were not significant (*p* < 0.001). The third model tested 21.4% of the sample. PA engagement and BMI showed a negative correlation (*p* < 0.001). People with higher BMI levels engaged less in PA. The most important predictors of PA engagement in the sample were self-efficacy (beta = 0.202) and BMI (beta = −0.055).

## 4. Discussion

### 4.1. PA Engagement in Older German Adults at Increased Dementia Risk with Multimorbidity

Established international recommendations suggest 5 days or 150 min of PA per week [13]. Only approximately half (51.8%) of our participants met this threshold. A relevant share of older German adults at increased dementia risk with multimorbidity, however, was not adequately physically active.

However, this study is the first to describe PA engagement, prevalence of PA types, stages of change for PA engagement, self-efficacy, and characteristics of participants according to the PA stages of change in community-dwelling persons with an increased risk of dementia and multimorbidity. In our study, 51.8% of participants reported being physically active at least twice a week for 30 min, which is consistent with the literature (32 to 70%) [51,52]. The most commonly performed PA types by participants were gardening and house chores, walking >10 min, cycling, and gymnastics, which is in line with previous findings [53,54,55].

According to the TTM stages of change for PA, the majority (46%) of study participants were in the action stage. A previous meta-analysis [56] also found that the majority of participants were in a physically active stage (maintenance) when applying similar criteria for PA (30 min for three days each week). Notably, international guidelines recommend at least moderate-intensity aerobic PA for 150 min/week [13]. The same systematic review [56] also showed comparable results for both the proportion of participants with no PA and no intention of engaging PA in the short term and for participants with the intention to begin PA in the short term.

Distinguishing different stages of change may allow individual matching of cognitive, affective, and behavioral techniques and strategies that can help individuals progress through the stages [30]. For example, individuals in the precontemplation stage might benefit from consciousness-raising development, which could be a strategy to guide them to the contemplation stage [57]. Therefore, consciousness-raising aims to encourage individuals to increase their knowledge, become aware of risks, care about consequences for others, and understand benefits. It therefore may help guide an individual move from having no intentions of being active to considering and intending to be active in the future [30]. Regular assessment and evaluation of patients’ perceived stages of change may be helpful for assessing interventions’ effects since some interventions might not lead to long-term behavior change directly but facilitate part of the change. By moving from precontemplation to contemplation, an individual does not become physically active, but he or she may have increased knowledge, perceptions of risk, and recognition of benefits.

Regarding the TTM stages of change, both inactive and active participants were similar in terms of sociodemographic characteristics, ADLs, alcohol consumption, and multimorbidity. The only significant group difference was found in participants’ BMI. Participants with a higher BMI ≥30 kg/m^2^ were less active (*p* = 0.001). Participants who lived with a partner and those who had a healthier diet more often were in the maintenance or action stage, although the findings were not significant. These results are consistent with previous studies, particularly in terms of BMI in more physically active persons [58].

### 4.2. Potential PA Determinants in Older German Adults at Increased Dementia RISK with Multimorbidity

#### 4.2.1. Psychological Factors

Most of the participants showed high levels of self-efficacy, as they felt confident in exercising even if they were worried (77.1%), felt tense (72.8%), or felt low (67.8%).

Self-efficacy seems to be an important predictor of PA participation among persons at risk of dementia and multimorbidity (*p* < 0.001). Previous studies have shown that enhancing self-efficacy might be linked to greater exercise willingness, and it might grow with each stage of change [56]. Self-efficacy influences the level of effort in which people seek to achieve their goals and the level of persistence despite difficulties or failures [59]. Accordingly, it can be argued that self-efficacy might generally play an important role in changing behaviors in persons with multimorbidity and the risk of developing dementia [57].

#### 4.2.2. Health-Related Factors

We were able to identify BMI as a potential health-related determinant. People with higher BMI levels participated less in PA (*p* < 0.001). This finding is consistent with results from a longitudinal study in older adults with 10-year follow-up [60]. Moreover, higher BMI negatively correlated with an increased decline in walking endurance [60]. Mikkola et al. [60] discussed different explanations for these outcomes. Fat mass can be understood as an additional mechanical load on the body that can affect people’s movement performance [60]. Similarly, obesity (BMI over 30 [61]) has been reported to decrease PA. Participants with high obesity were found to be less physically active [60]. Similarly, fat mass possibly interferes with physical performance due to certain mechanisms, such as inflammation, atherosclerosis, and insulin resistance. Such mechanisms can also lead to physical performance impairment, such as reduced cardiovascular function, slowed walking, lower endurance, and impaired muscle strength [60].

### 4.3. Limitations

Our study has some limitations, including self-reported PA validity and the assessment of PA as a dichotomous variable (PA for at least two times per week for at least 30 min, yes or no). Thus, we did not evaluate individual quantitative dimensions of PA, such as individual frequency, duration, and intensity of PA. Such information would have provided us with more precise data to determine the level of PA engagement in the population studied. Moreover, the cross-sectional design did not allow us to determine causality. The results of this study are limited to community-dwelling elderly Germans with a high risk of dementia and multimorbidity.

## 5. Conclusions

The AgeWell.de study, the first lifestyle trial to prevent cognitive decline in Germany, contributes relevant evidence to the growing field of dementia prevention based on risk factor modifications, particularly physical inactivity. Although the present study is limited, it provides an initial approximation to account for PA engagement and possible PA determinants in elderly adults with multimorbidity and risk of dementia. Consequently, based on the regression model presented, our findings suggest that self-efficacy and BMI may be determinants that influence PA engagement in persons with multimorbidity and risk of developing dementia. Thus, developers of lifestyle interventions should consider these factors when planning and implementing initiatives to promote PA among inactive adults with multimorbidity as well as assess stages of change to identify intervention effects. Increased confidence and PA self-efficacy may have positive effects on initiating and maintaining PA, as self-efficacy facilitates information sources, including performance accomplishments, vicarious learning, verbal encouragement, and physiological and effective states [57].

## Figures and Tables

**Figure 1 ijerph-19-03164-f001:**
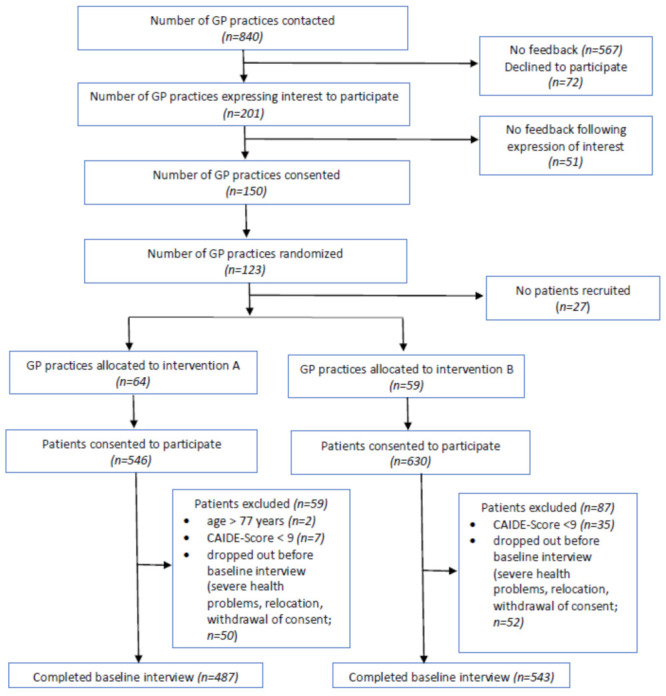
Recruitment process and sample selection of participants in the AgeWell.de trial.

**Table 1 ijerph-19-03164-t001:** Baseline characteristics of study participants in the AgeWell.de trial.

Variable	*n*	Total Sample *n* = 1.030	Men *n* = 493	Women *n* = 537	*p*-Value
**Sociodemographic characteristics**					
Age in years, mean (SD)	1.030	69.0 (4.9)	69.0 (4.9)	69.2 (4.9)	0.078 ^b^
Education (CASMIN)					
Low, *n* (%)	1.030	248 (24.1)	129 (12.5)	119 (11.6)	
Intermediate, *n* (%)		569 (55.2)	243 (23.6)	326 (31.7)	
High, *n* (%)		213 (20.7)	121 (11.7)	92 (8.9)	0.001 ^a^
Income in euros, mean (SD)	930	1572.2 (845.8)	1615.9 (901.3)	1531.8 (789.9)	0.129 ^b^
Marital status					
Married/cohabitating, *n* (%)	1.030	665 (64.6)	382 (77.5)	283 (52.7)	
Unmarried/divorced/widowed, *n* (%)		365 (35.4)	111 (22.5)	254 (47.3)	0.000 ^a^
**Cognitive status**					
MoCA sum score points (0–30), mean (SD)	1026	24.5 (3.1)	24.7 (3.0)	24.4 (3.1)	0.102 ^b^
Normal (>26), *n* (%)		405 (39.5)	204 (19.9)	201 (19.6)	
Mild impairment (18–25), *n* (%)		592 (57.7)	275 (26.8)	317 (30.9)	
Moderate impairment (10–17), *n* (%)		28 (2.7)	11 (1.1)	17 (1.7)	
Severe impairment (<10), *n* (%)		1 (0.1)	0 (0.0)	1 (0.1)	0.357 ^a^
**GPs diagnoses**					
Depression, *n* (%)	1022	118 (11.5)	62 (6.0)	56 (5.4)	0.250 ^a^
Diabetes mellitus type 2, *n* (%)	1023	399 (38.7)	193 (18.7)	206 (20.0)	0.937 ^a^
History of myocardial infarction, *n* (%)	1016	60 (5.8)	35 (3.4)	25 (2.4)	0.108 ^a^
History of stroke, *n* (%)	1019	45 (4.4)	23 (2.2)	22 (2.1)	0.895 ^a^
Hypercholesterolemia/Hyperlipidemia, *n* (%)	1014	735 (71.4)	353 (34.3)	382 (37.1)	0.759 ^a^
Renal insufficiency/chronic kidney disease (%)	1012	161 (15.6)	83 (8.1)	78 (7.6)	0.461 ^a^
Hypertension, *n* (%)	1023	895 (87.0)	421 (40.9)	474 (46.0)	0.335 ^a^
Coronary heart disease, *n* (%)	1020	176 (17.1)	79 (7.7)	97 (9.4)	0.683 ^a^
Obesity (BMI ≥ 30 kg/m^2^) (%)	1023	556 (54.4)	266 (25.8)	290 (28.2)	0.454 ^a^
**Health and lifestyle factors**					
Geriatric Depression Scale (points), mean (SD)	1016	1.6 (2.0)	1.6 (2.0)	1.66 (2.1)	0.533 ^b^
Lubben Social Network Scale (points), mean (SD)	798	20.76 (6.5)	20.8 (6.6)	20.7 (6.4)	0.903 ^b^
Activities of daily living (Barthel Index points), mean (SD)	1028	99.5 (3.1)	99.5 (2.5)	99.4 (3.4)	0.441 ^b^
Current smoker, *n* (%)	1023	127 (12.3)	66 (6.5)	61 (6.0)	0.369 ^a^
Alcohol drinking ≥ once a week, *n* (%)	889	533 (60.0)	256 (28.8)	277 (31.2)	0.870 ^a^
Daily consumption of 5 portions of fruits and vegetables, *n* (%)	1023	497 (48.6)	225 (22.0)	272 (26.6)	0.238 ^a^

^a^ Pearson’s chi-squared test, ^b^ Welch’s t test, SD = standard deviation.

**Table 2 ijerph-19-03164-t002:** Baseline PA engagement of study participants in the AgeWell.de trial.

PA at Least 2 Times/Week for 30 min (Adapted from Lippke et al. [45]	*n*	Total Sample	Men	Women	*p*-Value
		*n* = 1.030	*n* = 493	*n* = 537	
YES, *n* (%)	1014	525 (51.8)	237 (23.4)	288 (28.4)	
NO, *n* (%)	1014	489(48.2)	243 (24.5)	237 (23.7)	0.066 ^a^
Stages of change for PA engagement based on TTM stages of change					
Precontemplation, *n* (%)		199 (20.0)	109 (22.7)	90 (18.8)	
Contemplation, *n* (%)		190 (19.0)	94 (19.6)	96 (20.0)	
Preparation, *n* (%)		91 (9.0)	40 (8.3)	51 (10.6)	
Action, *n* (%)		52 (5.1)	24 (4.6)	28 (5.3)	
Maintenance, *n* (%)		466 (46.0)	208 (39.6)	258 (49.1)	0.066 ^a^
Self-efficacy/Confident to exercise when feeling…					
Worried and have problems (yes), *n* (%)	1023	748 (77.1)	357 (76.9)	391 (77.3)	0.902 ^a^
Low (yes), *n* (%)		658 (67.8)	315 (68.0)	343 (67,7)	0.899 ^a^
Tense (yes), *n* (%)		704 (72.8)	345 (74.7)	359 (71.1)	0.211 ^a^
Tired (yes), *n* (%)		477 (49.2)	214 (46.1)	263 (52.0)	0.068 ^a^
Busy (yes), *n* (%)		438 (45.2)	197 (42.5)	241 (47.6)	0.112 ^a^

^a^ Pearson’s chi-squared test.

**Table 3 ijerph-19-03164-t003:** Prevalence of types of PA for the total sample and for men and women; in % (95%-CI).

Types of PA	*n*	Total Sample *n* = 1.030	Men *n* = 493	Women *n* = 537	*p*-Value
Cycling, % (95%-CI)	1022	1.22 (1.13;1.31)	1.34 (1.21;148)	1.11 (0.98;1.24)	0.013 ^a^
Walking > 10 min, % (95%-CI)	1022	2.43 (2.35;2.52)	2.30 (2.17;2.42)	2.56 (2.45;2.68)	0.002 ^a^
Swimming, % (95%-CI)	1020	0.45 (0.40;0.50)	0.41 (0.33;0.48)	0.49 (0.41; 0.57)	0.139 ^a^
Gymnastics, % (95%-CI)	1016	1.08 (1.00;1.17)	0.79 (0.67;0.91)	1.35 (1.23;1.48)	0.000 ^a^
Fitness, % (95%-CI)	1020	0.52 (0.45;0.59)	0.64 (0.53;0.74)	0.41 (0.32;0.50)	0.001 ^a^
Sports, % (95%-CI)	1020	0.08 (0.06;0.11)	0.13 (0.09;0.18)	0.04 (0.02;0.06)	0.000 ^a^
Gardening and house chores, % (95%-CI)	1011	3.11 (3.03;3.19)	3.00 (2.88;3.1)	3.22 (3.11;3.33)	0.006 ^a^

^a^ Pearson’s chi-squared test.

**Table 4 ijerph-19-03164-t004:** Sociodemographic and health-related characteristics according to participants’ stages of PA engagement.

Variable	Patients in the Precontemplation, Contemplation or Preparation PA Stage *n* = 480	Patients in the Maintenance or Action PA Stage *n* = 518	*p* Value
Age, mean (SD)	68.8 (4.9)	69.0 (4.9)	0.571 ^b^
Intermediate education level, *n* (%)	283 (59.0)	278 (53.6)	0.011 ^a^
Income in euros, mean (SD)	1588.6 (871.2)	1555.9 (831.6)	0.561 ^b^
Living with partner, *n* (%)	304 (63.3)	351 (67.8)	0.334 ^a^
MoCa score, mean (SD)	24.5 (3.0)	24.5 (3.1)	0.750 ^b^
BMI ≥ 30 kg/m^2^, *n* (%)	290 (60.4)	254 (49.0)	0.001 ^a^
Lubben Social Network Scale, mean (SD)	20.8 (6.3)	20.7 (6.7)	0.731 ^b^
Barthel Index, points), mean (SD)	99.4 (3.0)	99.5 (3.1)	0.889 ^b^
Current smoker, *n* (%)	57 (11.8)	69 (13.3)	0.148 ^a^
Alcohol drinking ≥ once a week, *n* (%)	254 (52.9)	273 (52.7)	0.854 ^a^
Daily consumption of 5 portions of fruits and vegetables, *n* (%)	227 (47.3)	262 (50.1)	0.561 ^a^
Depressive symptoms, *n* (%)	59 (12.3)	58 (11.2)	0.737 ^a^
Diabetes mellitus Type 2, *n* (%)	178 (37.0)	218 (42.0)	0.132 ^a^
History of myocardial infarction, *n* (%)	24 (5.0)	35 (6.8)	0.303 ^a^
History of stroke, *n* (%)	23 (4.8)	22 (4.2)	0.842 ^a^
Hypercholesterolemia /hyperlipidemia, *n* (%)	361 (75.2)	361 (69.7)	0.122 ^a^
Renal insufficiency/chronic kidney disease (%)	73 (15.2)	87 (16.8)	0.286 ^a^
Hypertension, *n* (%)	422 (87.9)	460 (88.8)	0.397 ^a^
Coronary heart disease, *n* (%)	78 (16.25)	96 (18.5)	0.480 ^a^

^a^ Pearson’s chi-squared test, ^b^ Welch’s t test.

**Table 5 ijerph-19-03164-t005:** Regression models for potential PA-related determinants.

Baseline Variables	PA at Least 2 Times/Week for 30 min								
	Model 1			Model 2			Model 3		
	B	95% CI	*p* Value	B	95% CI	*p* Value	B	95% CI	*p* Value
Self-efficacy	0.202	1.174;1.276	0.000	0.204	1.176;1.279	0.000 *	0.200	1.171;1.274	0.000 *
Cognitive status	−0.031	0.915;1.027	0.295	−0.020	0.924;1.041	0.518	−0.013	0.928;1.049	0.676
Sex				0.258	0.892;1.878	0.175	0.291	0.916;1.954	0.132
Age				0.026	0.989;1.064	0.165	0.013	0.975;1.052	0.502
Educational level				0.108	0.971;1.277	0.124	0.091	0.953;1.260	0.200
Income				0.000	1.000;1.000	0.160	0.000	1.000;1.000	0.417
Marital status				−0.071	0.850;1.021	0.131	−0.058	0.860;1.036	0.225
BMI							−0.055	0.913;0.981	0.002 *
Depression							0.001	0.907;1.105	0.982
Social network							−0.006	0.964;1.025	0.704
ADLs							−0.004	0.937;1.059	0.908
Current smoker							0.100	0.887;1.379	0.372
Alcohol consumption (≥once a week)							−0.081	0.843;1.008	0.076
Fruits and vegetables consumption (5 daily portions)							−0.021	0.713;1.344	0.895
Constant	−2.191			−4.508			−1.555		
R^2^	0.183			0.196			0.214		

* *p* < 0.05; Model 1 = psychological determinants; Model 2 = sociodemographic determinants; Model 3 = health-related determinants.

## Data Availability

The data presented in this study are available on request from the corresponding author. The data are not yet publicly available, as the trial is currently ongoing.

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
