# Peer review of "Physical Activity Determinants in Older German Adults at Increased Dementia Risk with Multimorbidity: Baseline Results of the AgeWell.de Study"

_ijerph, 2022, doi:10.3390/ijerph19063164_

Round 1

Reviewer 1 Report

The Manuscript (ijerph-1604769, title “Physical Activity Determinants in Older German Adults at Increased Dementia Risk with Multimorbidity: Baseline Results of the AgeWell.de Study-A Pragmatic Cluster-Randomized Controlled Lifestyle Trial”) aimed to provide a descriptive analysis of sociodemographic, health-related lifestyle factors and physical activity engagement in the German population. In general, the Manuscript is interesting, but is not well-written and this research showed a low level of novelty. The major limitation of this study is the use of self-reported questionnaires for the evaluation of the physical activity without any objective individual evaluation of its duration and intensity.

    Specific comments:

Line 86 Authors should delete odds ratio (OR) and confidence interval (CI) in the Introduction because these results are not important in this sentence.

Line 97 Authors indicated "other studies" but reported only one reference. Please indicate other references as regards obesity and physical activity.

Line 101 Authors should delete odds ratio (OR), confidence interval (CI) and significance level in the description of the Introduction since it is not important in this section.

Lines 133- 139 The aim of the study should be indicated at the end of Introduction section.

Line 149 It is not clear if the inclusion criteria are considered only an age of 60 and 77 years or from 60 to 77 years.

Line 232 The number of women and men enrolled should be similar. This finding could make a bias in the study.

Line 232 It should be “1.030…”.

Line 259 "Table 95" probably there is same mistake. It should be “table 3”.

Line 259-260 The sentence should be put into discursive form and not as a list of data with confidence intervals.

Line 278 “Table 5…” please delete the full stop after table 5.

Line 351 The reference of Mikkola et al., in the Discussion should be indicated with the relative number in the square brackets.

Author Response

Response to Reviewer 1 Comments

Thank you for your review of our paper. Please find our point-to point reply below.

Point 1: In general, the Manuscript is interesting but is not well-written and this research showed a low level of novelty.

Response 1: To ensure that the manuscript was well written, professionals reviewed and edited it. Please see attached data for certification of the editing.

In addition, we believe that the present study demonstrates novelty since, despite its limitations, it highlights physical activity determinants for the initiation and maintenance of physical activity in people with multimorbidity and at dementia risk. A topic that has not been sufficiently investigated in the current literature. It also provides the basis for developing effective treatment strategies, including physical exercise tailored towards the need of individual patients.

Point 2: The major limitation of this study is the use of self-reported questionnaires for the evaluation of the physical activity without any objective individual evaluation of its duration and intensity

Response 2: To date, there is no standard method of measuring physical activity using questionnaires in persons at risk of dementia and multimorbidity. However, most published studies assessing physical activity in persons with cognitive decline have administrated self-report questionnaires. Based on this information, we decided on using questionnaires that had been developed or previously validated for use with older adults. We also believe that there is the potential to develop and validate self-report physical activity questionnaires for persons at risk of dementia and multimorbidity, so long that it is developed in a manner best suited for use within this population.

In addition, considering that this assessment was conducted within a large project, there were other variables to assess. So we had to make compromises in terms of complexity and extensiveness so that the participants would not become exhausted and would actually perform the evaluations of all the variables included.   

Point 3: Line 86 Authors should delete odds ratio (OR) and confidence interval (CI) in the Introduction because these results are not important in this sentence.

Response 3: We agreed with the reviewer on this important remark. Therefore, we deleted odds ratio (OR) and confidence interval (CI) in the introduction section (Line 86).

Point 4: Line 97 Authors indicated "other studies" but reported only one reference. Please indicate other references as regards obesity and physical activity.

Response 4: Thank you for your suggestion. We indicated other references regarded obesity and physical activity, lower education, marital status and current smoking (Line 100). The new references are presented below:

Obesity: Cárdenas Fuentes, G., et al., Association of physical activity with body mass index, waist circumference and incidence of obesity in older adults. Eur J Public Health, 2018. 28(5): p. 944-950.

Lower education: Shaw, B.A. and L.S. Spokane, Examining the association between education level and physical activity changes during early old age. J Aging Health, 2008. 20(7): p. 767-87.

Marital status: Pettee, K.K., et al., Influence of marital status on physical activity levels among older adults. Med Sci Sports Exerc, 2006. 38(3): p. 541-6.

Current smoking: Swan, J.H., et al., Smoking Predicting Physical Activity in an Aging America. J Nutr Health Aging, 2018. 22(4): p. 476-482.

Point 5: Line 101 Authors should delete odds ratio (OR), confidence interval (CI) and significance level in the description of the Introduction since it is not important in this section.

Response 5: The odds ratio (OR), confidence interval (CI) and significance level were deleted from line 101 as recommended by the reviewer. Thank you for this suggestion.

Point 6: Lines 133- 139 The aim of the study should be indicated at the end of Introduction section.

Response 6: Thank you for your suggestion. As you mention, the objective of the study is indicated at the end of the introduction, precisely in line 125.

Point 7: Line 149 It is not clear if the inclusion criteria are considered only an age of 60 and 77 years or from 60 to 77 years.

Response 7: Thank you for your remark. We agree that this inclusion criterion is not clear. Therefore, we have modified the text as follows:

Lines 150 and 151: "between 60 to 77 years".

Point 8: Line 232 The number of women and men enrolled should be similar. This finding could make a bias in the study.

Response 8:

Point 9: Line 232 It should be “1.030…”.

Response 9: Thank you for your comment, we have modified the text as shown below:

Line 233: “1.030”

Point 10: Line 259 "Table 95" probably there is same mistake. It should be “table 3”.

Response 10: Thank you for this valuable observation. Unfortunately, as you point out, it was a mistake, and it was corrected in the manuscript.

Line 260: “Table 3”.

Point 11: Line 259-260 The sentence should be put into discursive form and not as a list of data with confidence intervals.

Response 11: The sentence was modified in the manuscript as follows:

Line 260: “The most frequent types of PA performed by the participants were gardening and house chores…”

Point 12: Line 278 “Table 5…” please delete the full stop after table 5.

Response 12: Thank you for your recommendation. The period after table 5 has been deleted as suggested.

Line 280: “Table 5...”

Point 13: Line 351 The reference of Mikkola et al., in the Discussion should be indicated with the relative number in the square brackets.

Response 13: The Mikkola et al. reference was added to the discussion (Line 354).

Reviewer 2 Report

Please find attachment.

Author Response

Response to Reviewer 2 Comments

Thank you for your review of our paper. Please find our point-to-point reply below.

Point 1: Title: the title is very bulky and might discourage people from reading. Therefore, it would be

beneficial to shorten it and make it more appealing, for examüle: Physical Activity Determinants in Older German Adults at Increased Dementia Risk with Multimorbidity: Baseline Results of the AgeWell.de-Study

Response 1: Thank you for this important remark. We have therefore modified the title to the one suggested by the reviewer.

“Physical Activity Determinants in Older German Adults at In-creased Dementia Risk with Multimorbidity: Baseline Results of the AgeWell.de Study”.

Point 2: Materials and methods: Line 149: “The inclusion criteria were age of 60 and 77. “Do you mean “age between 60 and 77” here? Please revise this sentence accordingly.

Response 2: Thank you for your remark. We agree that this inclusion criterion is not clear. Therefore, we have modified the text as follows:

Lines 150 and 151: “between 60 to 77 years”.

Point 3:  Table 5: In the table legend, it is stated that p values below 0.05 are written in bold.

However, in the table, there are no bold values. Please write significant p values in bold.

Response 3: Thank you for your remark. We have marked in bold, within table 5, those p-values below 0.05.

Point 4: Strengths and limitations: This section only contains limitations so far. I would recommend either renaming this section to “limitations” only or adding some points of strengths.

Response 4: That section has been modified as "limitations" (line 361).

Reviewer 3 Report

I consider this to be a relevant study, especially with regard to the specific knowledge on PA determinants and stages of change in persons with risk of dementia and multi-morbidity.
This knowledge can be important in developing specific interventions directed at target groups.

Author Response

Response to Reviewer 3 Comments

Thank you for your reviewing the manuscript " Physical Activity Determinants in Older German Adults at In-creased Dementia Risk with Multimorbidity: Baseline Results of the AgeWell.de Study ".

We have carefully revised the comments and have adapted the manuscript accordingly (the changes can be seen in the manuscript in red color). We believe that the manuscript improved and addressed the reviewers and now conveys the relevant information properly.

We also appreciate your positive feedback in which you find our study relevant and of scientific value. 

Round 2

Reviewer 1 Report

Authors should delete the odds ratio (OR) and confidence interval (CI) in the introduction section (Line 86).

In the revised draft of the Manuscript this finding is not revised.

Author Response

Response to Reviewer 1 Comments

Thank you for reviewing our paper once again. Please find our point-to-point reply below.

Point 1: Moderate English changes required

Response 1: Thank you for this suggestion. The manuscript has been modified using for a second time editing services. Please find the changes in the manuscript with Microsoft Word Track Changes.

Also please find attached the new certificate of the editing service.

Point 2: Authors should delete the odds ratio (OR) and confidence interval (CI) in the introduction section (Line 86). In the revised draft of the Manuscript this finding is not revised.

Response 2: Thank you for this important remark. The manuscript has been revised and the odds ratio (OR) and confidence interval (CI) were deleted from the introduction section (Line 87).
